# ZZ-dependent regulation of p62/SQSTM1 in autophagy

Yi Zhang [1], Su Ran Mun [2], Juan F. Linares[3], JaeWoo Ahn[1], Christina G. Towers[1], Chang Hoon Ji[2], Brent E. Fitzwalter[1], Michael R. Holden[1], Wenyi Mi [4], Xiaobing Shi [4], Jorge Moscat[3], Andrew Thorburn[1], Maria T. Diaz-Meco[3], Yong Tae Kwon[2,5] & Tatiana G. Kutateladze [1]

Autophagic receptor p62 is a critical mediator of cell detoxification, stress response, and metabolic programs and is commonly deregulated in human diseases. The diverse functions of p62 arise from its ability to interact with a large set of ligands, such as arginylated (Nt-R) substrates. Here, we describe the structural mechanism for selective recognition of Nt-R by the ZZ domain of p62 (p62$_{ZZ}$). We show that binding of p62$_{ZZ}$ to Nt-R substrates stimulates p62 aggregation and macroautophagy and is required for autophagic targeting of p62. p62 is essential for mTORC1 activation in response to arginine, but it is not a direct sensor of free arginine in the mTORC1 pathway. We identified a regulatory linker (RL) region in p62 that binds p62$_{ZZ}$ in vitro and may modulate p62 function. Our findings shed new light on the mechanistic and functional significance of the major cytosolic adaptor protein p62 in two fundamental signaling pathways.

[1] Department of Pharmacology, University of Colorado School of Medicine, Aurora, CO 80045, USA. [2] Protein Metabolism Medical Research Center and Department of Biomedical Sciences, College of Medicine, Seoul National University, Seoul 03080, Republic of Korea. [3] Cancer Metabolism and Signaling Networks Program, Sanford Burnham Prebys Medical Discovery Institute, La Jolla, CA 92037, USA. [4] Center for Epigenetics, Van Andel Research Institute, Grand Rapids, MI 49503, USA. [5] Protech Inc., Yongeon 103 Daehangno, Jongno-gu, Seoul 110-799, Republic of Korea. These authors contributed equally: Yi Zhang, Su Ran Mun. Correspondence and requests for materials should be addressed to Y.T.K. (email: yok5@snu.ac.kr) or to T.G.K. (email: tatiana.kutateladze@ucdenver.edu)

Sequestosome 1 (SQSTM1 or p62) mediates cell proliferation, survival, and death through multiple signaling programs, including autophagy and metabolism. This versatile protein adaptor functions as a signaling hub capable of recruiting diverse binding partners and is known to be misregulated in cancer and neurodegenerative disorders[1–5]. New studies of selective protein degradation reveal a critical role of p62 in the autophagic proteolytic cascade responsible for sequestration of toxic, aggregate-prone cargo proteins[6–8]. Following binding to the ubiquitinated cargos, p62 undergoes oligomerization and delivers the cargo aggregates to the autophagosome via interacting with the autophagosomal membrane protein LC3. The sequestered cargo is subsequently degraded by lysosomal enzymes when the autophagosome fuses with a lysosome.

The autophagosomal degradation pathway involves the N-end rule-dependent sensing of a degradation signal, named N-degron[9,10]. The primary determinant of N-degron is a desta-bilizing amino-terminal residue that can be produced via proteolytic cleavage of the substrate or enzymatically added and is targeted by a protein effector, or N-recognin[9,11]. One of the major and widespread N-degrons in eukaryotes is the amino-terminal arginine residue (Nt-R). Arg-tRNA transferases enzymatically add arginine to the protein sequence starting with either aspartic acid or glutamic acid and the resultant arginylated sequence is recognized by the UBR-box domain of UBRs[12–15]. Recent studies have identified the ZZ domain of p62 (p62$_{ZZ}$) as a new Nt-R recognin[6–8], however, the molecular mechanism by which it interacts with the degron and the biological importance of this interaction remain unclear.

p62 is also an important modulator of the nutrient sensor mTORC1[16,17]. It functions as a scaffold protein that recruits components of the mTORC1 signaling machinery to a specific subcellular location[18]. Free amino acids, lysine and arginine particularly, activate the mTORC1 response and promote p62 phosphorylation[19]. Binding of p62 to the E3-ubiquitin ligase TRAF6 then leads to polyubiquitination of the mTORC1 complex subunit and facilitates its translocation to the lysosome[18,19].

Structurally, p62 contains six functional motifs: an N-terminal PKC-binding PB1 (Phox and Bem1p) domain, the central ZZ and TB modules, a LC3-interacting region (LIR), a Keap1-binding region (KIR), and the C-terminal ubiquitin associated (UBA) domain (Fig. 1a). Various cellular stresses have been shown to stimulate p62 oligomerization, mediated by the PB1 domain and the region C-terminal to PB1[8,20,21]. The PB1-mediated oligomerization is essential for cargo collection and aggregation by p62 and the delivery of p62-cargo complexes to the autophagosome[6,21].

Here, we report the structural basis underlying specific recognition of Nt-R substrates by the ZZ domain of p62. We demonstrate that this interaction promotes p62 aggregation and is necessary for macroautophagy. We propose a novel mechanism for p62 autoregulation in which binding of the ZZ domain to the internal p62 sequence may play a role in regulating p62 function.

## Results and Discussion

**Structural basis for the Nt-R recognition by p62$_{ZZ}$.** To gain insight into the molecular mechanism by which p62 recognizes Nt-R substrates, we determined the atomic-resolution structure of the p62$_{ZZ}$-RE complex. We designed and crystallized a chimeric construct containing the Arg-Glu sequence fused via a short linker to the N-terminus of the ZZ domain of p62 (aa 120–171) (hereafter referred to as RE-ZZ) (Figs. 1b, c). In the structure, two RE-ZZ molecules form a complex in which the RE residues of one molecule are bound by the ZZ domain of another molecule, whereas the linker residues are flexible and

have no contact with either ZZ domain (Supplementary Figure 1). An extensive network of salt bridges and hydrogen bonds stabilizes the p62$_{ZZ}$-RE complex (Figs. 1b, c). The Nt-R is bound in a highly negatively charged groove lined with the residues D129, N132, D147, and D149 of p62. The guanidinium moiety of Arg1 is constrained by a salt bridge with the carboxyl group of D129 and hydrogen bonds with the oxygen atom of the carboxamide group of N132. Notably, the α-amino-terminal NH$_3^+$ group of Arg1 is bound via a set of intermolecular contacts, including salt bridges and hydrogen bonds with the carboxyl groups of D129 and D149 and the backbone carbonyl group of I127. In addition, the carbonyl group of Arg1 is hydrogen bonded to the backbone amide of I127. The complex is further stabilized through the interactions involving degron's Glu2. The side chain carboxyl group of Glu2 forms a salt bridge with the guanidinium moiety of R139, which in turn is restrained through electrostatic and hydrogen bonding contacts with D149, whereas the backbone amide of Glu2 donates a hydrogen bond to the carboxyl group of D147 (Fig. 1c). Remarkably, the same residues R139, D149, and D129 that are implicated in the intermolecular interactions with the Nt-R substrate, also interact with each other, yielding a unique arrangement of intertwined contacts that define the p62$_{ZZ}$ selectivity towards the RE sequence (Figs. 1c, 2a).

**The selectivity of p62$_{ZZ}$.** To establish the selectivity, we tested p62$_{ZZ}$ in $^1$H,$^{15}$N heteronuclear single quantum coherence (HSQC) titration experiments (Figs. 1d, e, 2b). As shown in Fig. 1d, the REEE peptide induced large chemical shift perturbations (CSPs) in uniformly $^{15}$N-labeled p62$_{ZZ}$. CSPs were in slow-to-intermediate exchange regime, indicative of a tight interaction. A number of amide crosspeaks of the p62$_{ZZ}$ apo-state disappeared upon addition of the peptide and another set of resonances corresponding to the bound state appeared (Fig. 1d). In contrast, no CSPs were detected when Ac-REEE peptide in which the α-amino-terminal NH$_3^+$ group of Arg1 is blocked by acetylation was titrated in, pointing to the critical role of the free amino terminus in the interaction (Fig. 1e, left panel). However, the free amino terminus alone was insufficient, as the binding was mostly eliminated when Arg1 was substituted with alanine in the AEEE peptide (Fig. 1e, middle panel). Titration of the RAEE peptide in which Glu2 is replaced with alanine led to CSPs that were in the intermediate, and no longer in the slow exchange regime, indicating a decrease in the binding activity of p62$_{ZZ}$ (Fig. 1e, right panel). In agreement, dissociation constant ($K_d$) measured by microscale thermophoresis (MST) for the interaction of p62$_{ZZ}$ with REEE was found to be 5.6 μM, whereas binding of the RAEE peptide was reduced ($K_d = 14$ μM) (Figs. 2c-e). The binding affinities were corroborated by intrinsic tyrosine fluorescence measurements (Figs. 2d, e). Altogether, these results indicate that an N-terminally unmodified arginine in the first position is required for the degron sequence to be recognized by p62$_{ZZ}$, and the presence of a glutamic acid next to the arginine further enhances this interaction.

A contribution of the RE-binding site residues to the binding energetics was assessed through mutating D129, D147, and D149 of p62$_{ZZ}$ individually to lysine and examining the mutant proteins by nuclear magnetic resonance (NMR) (Fig. 2b). The absence of CSPs in NMR samples of the three mutants upon titration with REEE indicated that each of these p62$_{ZZ}$ residues is required for the interaction. Furthermore, binding capabilities of N132A and R139A mutants of p62$_{ZZ}$ were also substantially compromised, implying that proper coordination of the Arg1 and Glu2 side chains is essential (Fig. 2e and Supplementary Figure 2).

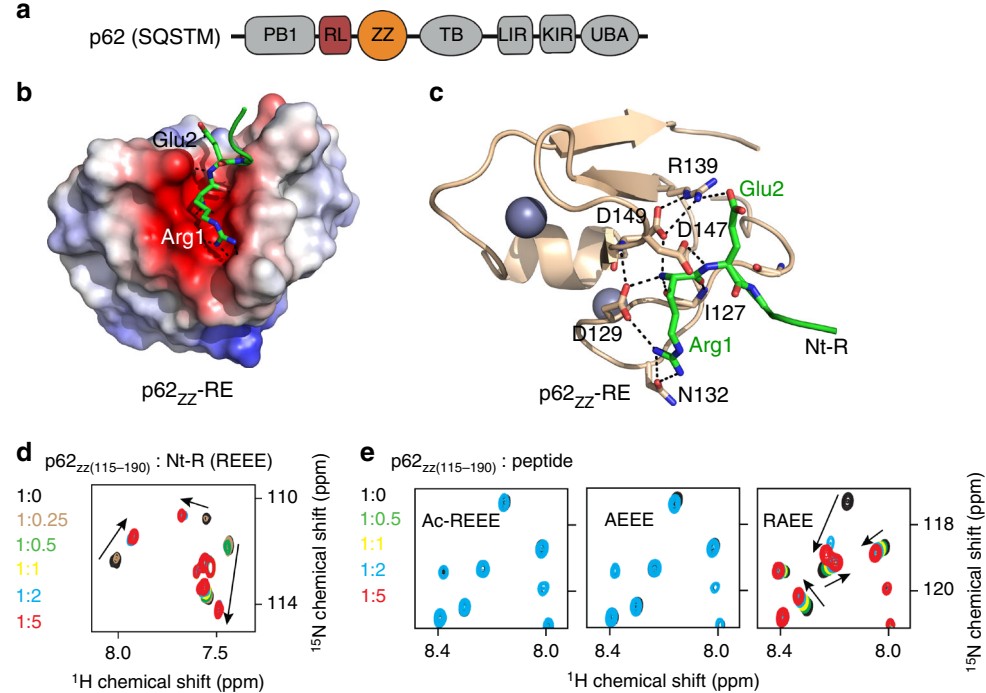

**Fig. 1** p62zz is a recognin of Nt-R. **a** p62/SQSTM domain architecture. **b** Electrostatic surface potential of p62$_{ZZ}$ colored blue and red for positive and negative charges, respectively. The bound Nt-R substrate (residues RE) are shown in stick. **c** A ribbon diagram of the crystal structure of p62$_{ZZ}$ (wheat) in complex with the Nt-R substrate (green). Dashed lines indicate hydrogen bonds and salt bridges. **d** Superimposed $^1$H,$^{15}$N HSQC spectra of p62$_{ZZ\ (115-190)}$ collected while the REEE peptide was titrated in. Spectra are color coded according to the protein:peptide molar ratio. **e** Superimposed $^1$H,$^{15}$N HSQC spectra of p62$_{ZZ\ (115-190)}$ collected upon titration with the indicated 4-mer peptides. Spectra are color coded according to the protein:peptide molar ratio

Much like the isolated p62$_{ZZ}$ domain, full-length p62 ectopically expressed in HEK293 cells robustly recognized arginylated substrates in pulldown assays. We tested X-nsP4 peptides, corresponding to the arginylated (X = Arg) or glyciny-lated (X = Gly) amino-terminal sequence of the N-end rule substrate Sindbis virus. Whereas wild-type p62 bound to the arginylated R-nsP4 peptide, it did not associate with the glycinylated G-nsP4 peptide (Fig. 2f). In contrast, none of the D129K, D147K, and D149K point mutant p62 proteins were able to recognize R-nsP4. Similarly, p62 associated with the peptides that correspond to the native substrates of p62—arginylated amino-terminal region of the ER-residing molecular chaperone BiP and arginylated amino-terminal region of CRT, but not with the respective peptides containing intrinsic glutamate or valine at the N-terminus (Figs. 2g, h).

To date, p62$_{ZZ}$ and the UBR-box are the only known receptors for type-I Nt-degron. Although both domains employ a set of negatively charged residues to engage the Nt-R degron, their structural topologies and the Nt-degron recognition mechanisms are different (Fig. 2i). Furthermore, p62$_{ZZ}$ and the UBR-box differ in their preferences for the residue at position 2 of the Nt-degron. Although p62$_{ZZ}$ prefers a negatively charged residue (Glu2, as in the physiological substrate BiP), the UBR domain shows preference for a hydrophobic residue due to a hydrophobic pocket located nearby[14]. This difference can help to explain the selective recognition and fine-tuned sorting of substrates toward proteasomal and autophagic degradation pathways.

**Binding of p62$_{ZZ}$ to arginylated substrates induces autophagy**. To define the biological significance of Nt-R recognition by p62$_{ZZ}$ in vivo, we monitored p62 puncta formation in p62-/- mouse embryonic fibroblast (MEF) cells, in which *p62$^{f/f}$* was deleted using the CRE recombinase, expressing WT protein and loss-of-

function mutant D129K. The cells were treated with or without XIE62-1004, a recently developed small molecule ligand of p62$_{ZZ}$ that induces p62 polymerization and facilitates collection of autophagic cargoes such as misfolded proteins and their aggre-gates[8]. The XIE62-1004 treatment readily induced formation of cytosolic puncta positive for p62 in MEF cells expressing WT p62, whereas cells expressing D129K mutant were incapable of responding to the treatment (Figs. 3a, b). We then examined XIE62-1004-induced autophagosome formation in cells express-ing WT p62 or the D129K, D147K and D149K mutants. After treatment with the autophagic blocker hydroxychloroquine (HCQ), immunoblotting analysis of LC3 showed that XIE62-1004 strongly promotes autophagosome biogenesis in cells expressing WT p62 (Fig. 3c). Immunoblotting assays also showed that XIE62-1004 promotes the lipidation of LC3 in WT MEFs, whereas p62-/- MEFs did not respond to XIE62-1004 (Fig. 3d). Furthermore, the efficacy of XIE62-1004 to induce LC3 lipidation was restored when recombinant p62 was stably expressed in p62-/- MEFs, but not when p62 mutants D129K, D147K, and D149K were expressed (Fig. 3d and Supplementary Figure 3a). Collectively, these data indicate that the binding of p62$_{ZZ}$ to arginylated substrates induces autophagy and is also required for p62 puncta formation in cells.

We further characterized autophagic function of p62$_{ZZ}$ through investigating different subtypes of autophagy, such as macro-autophagy and mitophagy. To quantitatively monitor functional autophagy, particularly autolysosome formation in cells, an mCherry-GFP-LC3 tandem construct was stably expressed, and the acidic quenching of green fluorescent protein (GFP) signal relative to the stable expression of mCherry was measured by flow cytometry[22] (Fig. 4a). In agreement with immunoblotting analysis of LC3, Earle's balanced salts solution (EBSS)-induced starvation caused a dramatic increase in the percent of cells undergoing

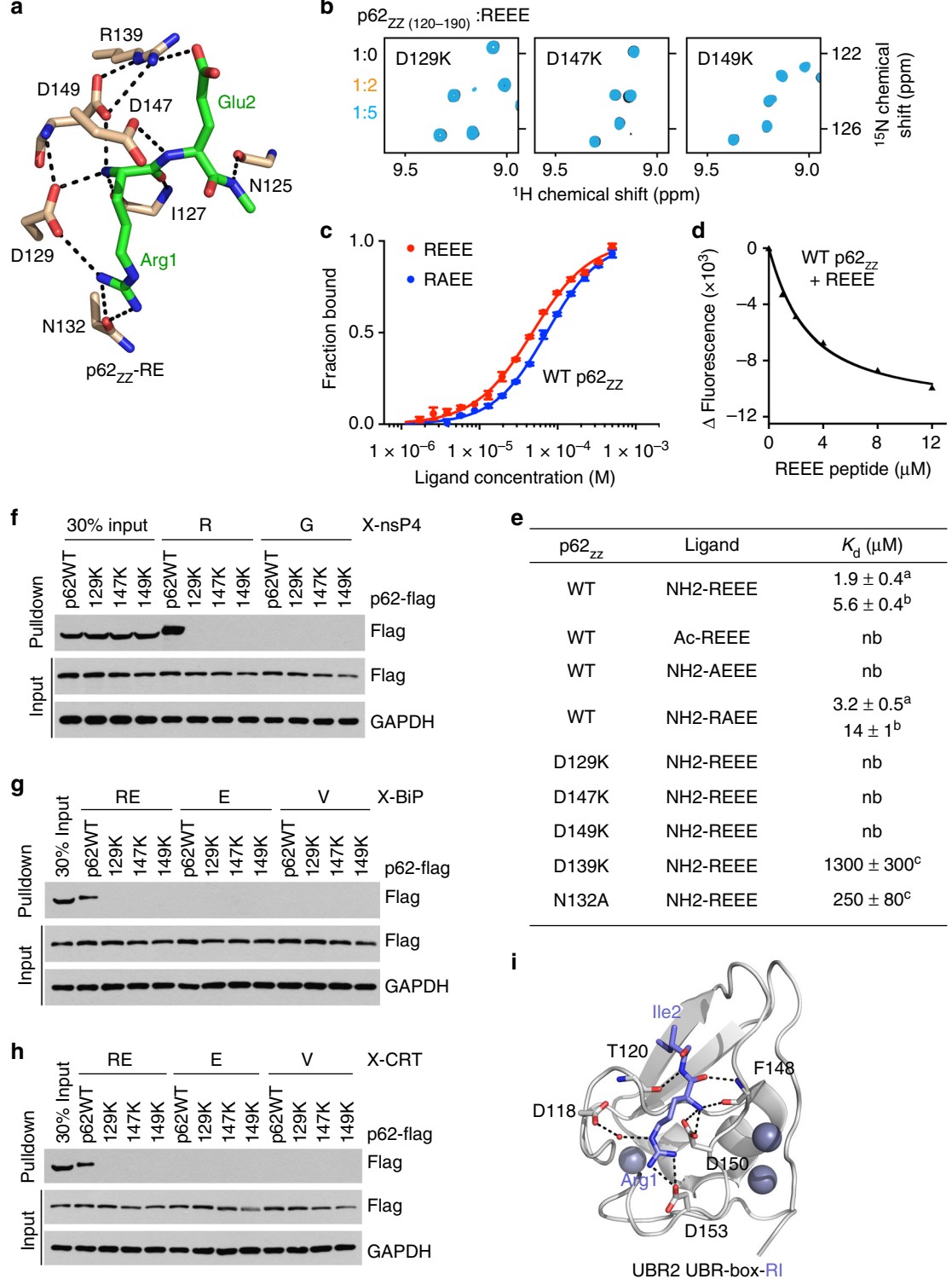

**Fig. 2** Molecular basis for the specific targeting of Nt-R by p62[ZZ]. **a** A zoom-in view of the Nt-R degron binding site. **b** Superimposed $^1$H,$^{15}$N HSQC spectra of p62$_{ZZ\ (120-190)}$ collected upon titration with the REEE peptide. Spectra are color coded according to the protein:peptide molar ratio. **c**, **d** Representative binding curves used to determine the $K_d$ values by MST (**c**) or tyrosine fluorescence (**d**). Error bar in **c** represents s.d. in triplicate measurements. **e** Binding affinities of p62$_{ZZ}$ for the indicated peptides measured by MST ([a], using p62$_{ZZ\ (115-190)}$) or tyrosine fluorescence ([b], using p62$_{ZZ\ (120-171)}$) or NMR ([c], using p62$_{ZZ\ (115-190)}$). Errors represent s.d. calculated from curve fitting (**a**, **c**) or duplicate midarguments (**b**). **f–h** Pulldown assays of wild-type or mutated p62-Flag expressed in HEK293 cells using X-nsP4 (**f**), X-BiP (**g**), and X-CRT (**h**) peptides. Uncropped blots are shown in Supplementary Figure 7. **i** A ribbon diagram of the structure of UBR-box of UBR2 (gray) in complex with Nt-R degron (blue, PDB 3NYN)

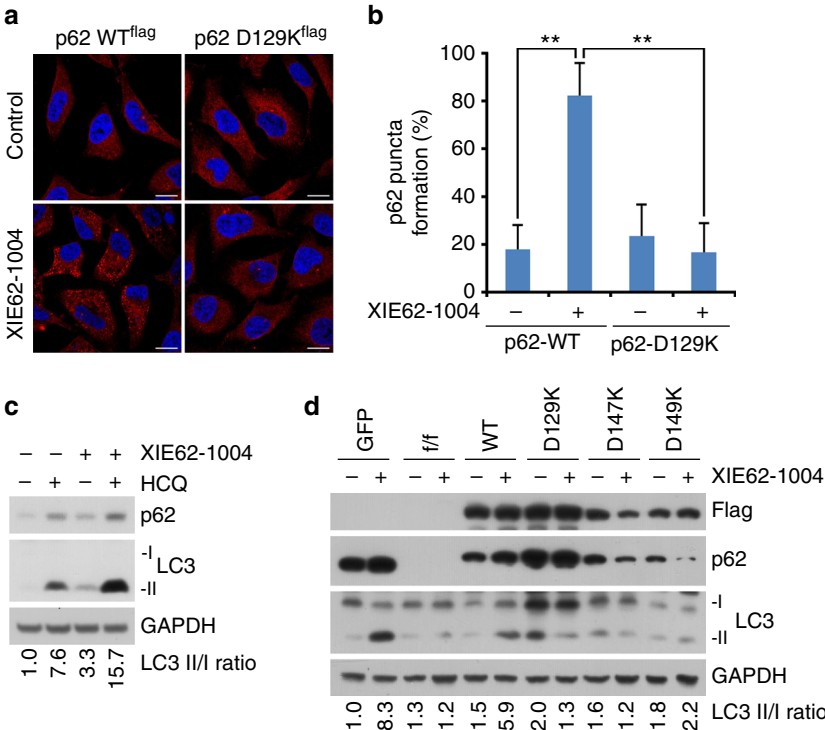

**Fig. 3** p62zz mediates stress-induced autophagy. **a** Mouse embryonic fibroblasts expressing recombinant wild-type or D129K point mutant p62-flag were treated with XIE62-1004 (5 µM, 6 h) or DMSO negative control, followed by immunostaining analysis. Scale bar, 10 µm. **b** Quantification of **a** (p calculated by two-tailed Student's t-test; data shown as mean ± S.D. of three independent experiments each with n = 30 cells; **p < 0.01). **c** Immunoblotting assay of endogenous p62-expressing mouse embryonic fibroblasts treated with XIE62-1004 (5 µM, 6 h) in the presence or absence of hydroxychloroquine (10 µM, 24 h) for autophagic flux assay. **d** Western blot analysis of mouse p62[fl/fl] embryonic fibroblasts infected with GFP or CRE adenoviruses were reconstituted with retroviruses expressing WT Flag-p62 or point mutants of Flag-p62 treated with/without XIE62-1004 (5 µM, 6 h). Ratio between LC3-II and LC3-I quantified by densitometry using ImageJ

autophagy, which was indicated by a decrease in GFP expression (Fig. 4b). In starved conditions, addback of WT p62 increased the percent of cells undergoing autophagy, however, rescue with D129K or D147K mutant of p62 failed to do so (Figs. 4c, d). Together, these results showed that induction of autophagy depends on the functional ZZ domain. Further analysis revealed that p62 affects macroautophagy but does not cause a significant increase in the percent of cells undergoing mitophagy, the form of autophagy that degrades mitochondria (Fig. 4e). Overall, these data suggest that p62zz is necessary for efficient macroautophagy induced by EBSS, whereas p62 is dispensable in mitophagy.

We next examined the role of p62zz in selective autophagy using cells under misfolded protein stress, induced by proteasome inhibitor MG132. Immunoblotting analysis showed that p62-/- MEFs failed to properly induce the lipidation of LC3 as compared with +/+ MEFs when the cells were treated with MG132 (Fig. 4f and Supplementary Figure 3b). Notably, stable expression of WT p62 allowed p62-/- MEFs to induce autophagy under the same condition (Fig. 4f). Consistent with the aforementioned results, mutation of either D129, D147, or D149 abolished the ability of cells to induce LC3 lipidation and autophagosome formation (Fig. 4g and Supplementary Figure 3b). These data point to the critical role of p62zz-Nt-R interaction in autophagy under proteasomal inhibition.

**p62zz mediates p62 aggregation in vitro**. p62-dependent selective autophagy involves polymerization of p62 in response to stress stimuli, including protein misfolding and oxidative stress[6,8,20,23,24], and in vitro, recombinant p62 forms aggregates within a few days[25]. Both the PB1 domain and the linker

connecting PB1 and ZZ have been implicated in polymerization[21,26,27]. PB1 was shown to form homo- and hetero-oligomers[23,26,27], whereas the linker region is needed to form disulfide-coupled aggregated conjugates via Cys113[8,20] and/or tubular structures, as demonstrated by cryo-EM studies[21]. To clarify the role of p62zz in p62 polymerization, we performed in vitro aggregation assays in which extracts from MEFs expressing p62 were treated with or without dipeptides and reactions were resolved by non-reducing sodium dodecyl sulfate–polyacrylamide gel electrophoresis (SDS-PAGE) (Supplementary Figure 4). In the absence of stimulation, WT p62 existed in a monomer-~250 kDa oligomers equilibrium, in support of the findings that this polymerization step is PB1 domain dependent[8] (Supplementary Figure 4a, b). The p62zz mutants (D129K, D147K, and D149K) also formed ~250 kDa oligomers, indicating that this step is ZZ domain independent. Consistent with previous reports[8], stimulation with the Arg-Ala peptide but not by the reverse Ala-Arg peptide induced p62 aggregation, which was disulfide bond formation dependent (Supplementary Figure 4a, c). However, the D129K, D147K, and D149K mutants of p62 that do not bind to Nt-R showed substantially reduced aggregation in the presence of either peptide, demonstrating that autophagic p62 aggregation depends on both the functional ZZ domain and stimulation with Arg-Ala (Supplementary Figure 4c). These results together with the autophagosome formation data suggest that functional p62zz is necessary for mediating autophagic aggregation of p62 in vitro and in vivo.

**p62 is not a direct arginine sensor in mTORC1 pathway**. To determine whether the amino-acid arginine, which is abundant

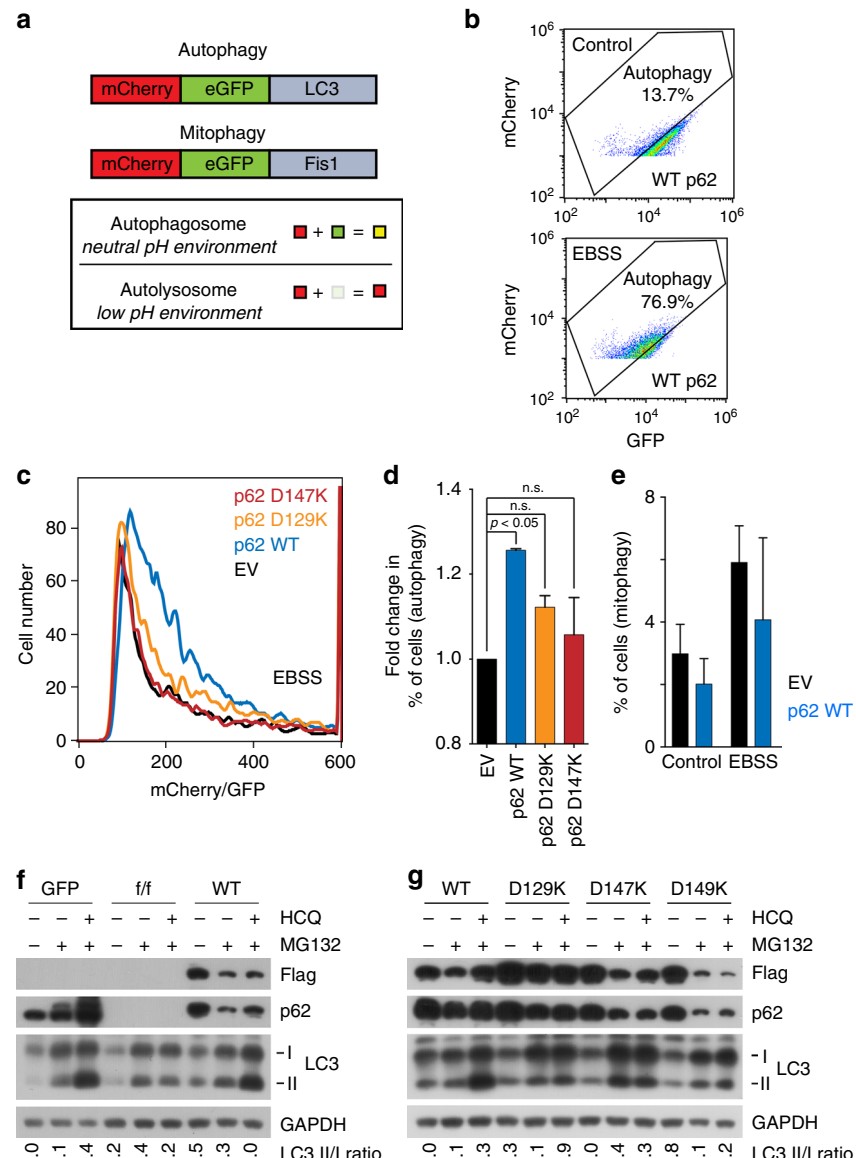

**Fig. 4** p62zz is necessary for efficient EBSS-induced macroautophagy. **a** Schematic of LC3 tandem and Fis-tandem flux assay. **b** An example of the LC3 tandem flow cytometry assay under basal conditions or starved (5% serum media in 95% EBSS media) conditions in p62 WT MEFs, where a decrease in GFP expression can be observed and quantified to measure the percent of cells undergoing autophagy. **c** Number of cells undergoing autophagy 24 h after starvation in 5% serum media in EBSS: measured by the decreased ratio of mCherry/GFP via flow cytometry in p62-/- MEFS with stable expression of an empty vector, WT p62, D129K p62 or D147K p62 and stable expression of mCherry-GFP-LC3 tandem (representative figure of two experiments). **d** Quantification of (**c**), where the percent of cells undergoing autophagy was normalized to the p62-/- condition. One-way ANOVA was performed on two independent experiments. **e** Normalized percent of cells undergoing mitophagy measured by the decreased ratio of mCherry/GFP via flow cytometry in p62-/- MEFS with stable expression of an empty vector, WT p62, D129K p62, or D147K p62 and stable expression of mCherry-GFP-Fis1 tandem. Error bar in **d** and **e** represents s.e.m with N of 2. **f** Western blot analysis of mouse p62^fl/fl embryonic fibroblasts infected with GFP or CRE adenoviruses were reconstituted with retroviruses expressing WT Flag-p62 or point mutants of Flag-p62 treated with/without MG132 (0.2 μM, 24 h) in the presence or absence of hydroxychloroquine (10 μM, 24 h). Ratio between LC3-II and LC3-I quantified by densitometry using ImageJ. **g** Similar to **f** but with mouse embryonic fibroblasts expressing recombinant wild-type p62-flag or indicated point mutants

in the cytosol, could be a substrate for p62$_{ZZ}$, we employed NMR titration experiments. As shown in Fig. 5a, a free arginine binds to p62$_{ZZ}$, but free glutamic acid, leucine, or glutamine do not. To elucidate the mechanistic basis for the arginine association, we obtained the crystal structure of p62$_{ZZ}$ in complex with arginine (Fig. 5b). Structural comparison of p62$_{ZZ}$-RE and p62$_{ZZ}$-arginine complexes reveals that the intermolecular contacts constraining the side chain and the α-amino-terminal NH$_3^+$ group of Arg1 are conserved in both complexes, however, D147 and R139 no longer contribute to the interaction with free arginine.

Given the ability to bind arginine and its central role as a positive regulator of the mTORC1 pathway, we reasoned that p62 could affect the capacity of mTORC1 to respond to arginine. Indeed, p62-/- cells, starved of arginine (Fig. 5c) or amino acids (Fig. 5d), showed a severe inhibition of the mTORC1 activation in response to arginine, suggesting that p62 is necessary for arginine-mediated mTORC1 activation. To determine whether the activation of mTORC1 by arginine requires the arginine-binding capacity of p62, we reconstituted p62-/- cells with retroviruses expressing p62 WT or the p62 mutants that are

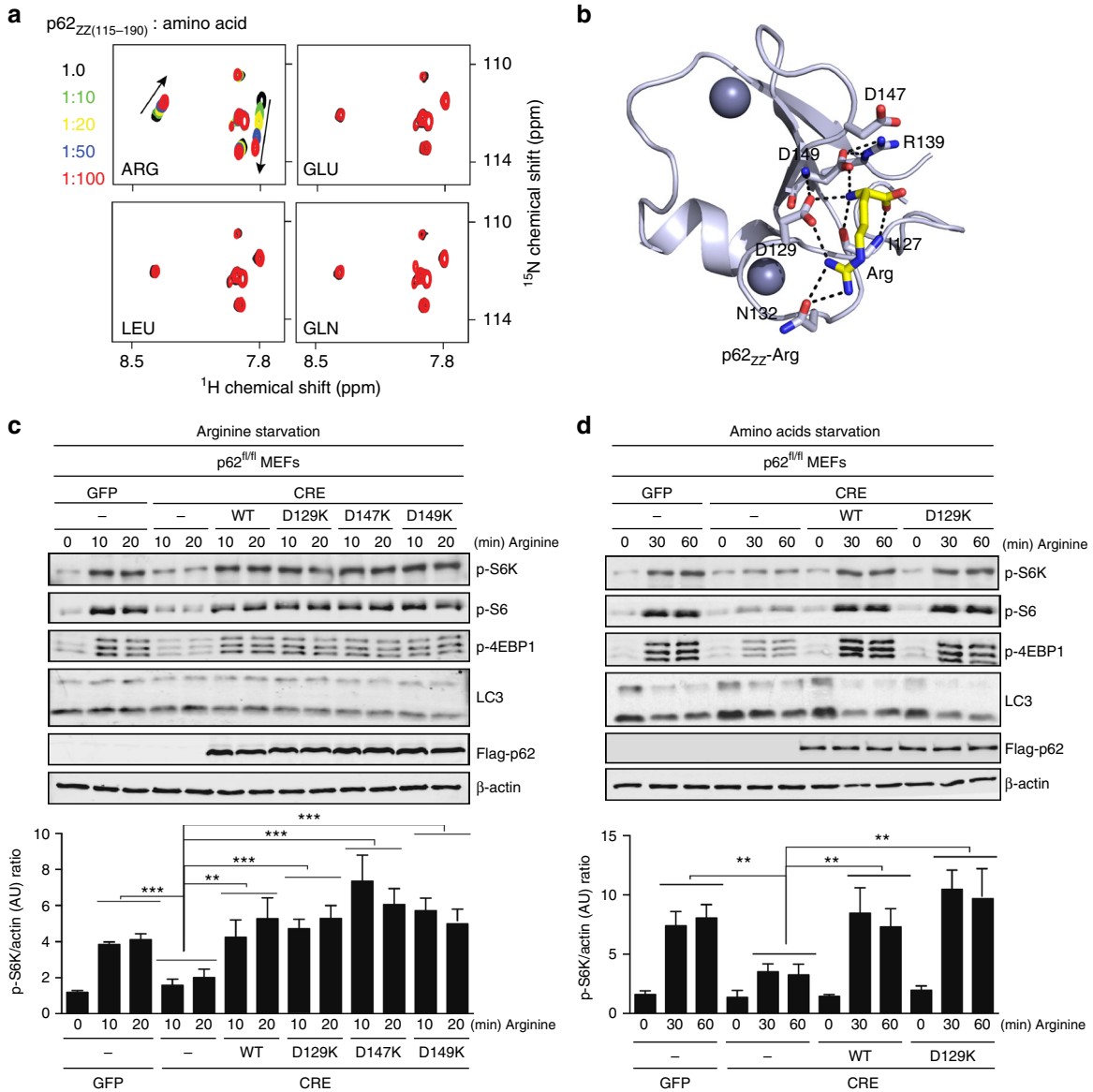

**Fig. 5** p62 is not a direct arginine sensor in mTORC1 pathway. **a** Superimposed $^1$H,$^{15}$N HSQC spectra of p62$_{ZZ\ (115–190)}$ collected upon titration with the indicated free amino acids. Spectra are color coded according to the protein:ligand molar ratio. **b** A ribbon diagram of the crystal structure of p62$_{ZZ}$ (wheat) in complex with the amino-acid arginine (yellow). **c** Mouse p62$^{fl/fl}$ embryonic fibroblasts infected with GFP or CRE adenoviruses were reconstituted with retroviruses expressing Flag-p62 WT, D129K, D147K, or D149K. Cells were deprived of arginine for 4 h and then stimulated with arginine for the indicated durations. Cell lysates were analyzed for the levels of the specified proteins. Graphs represent p-S6K/actin ratio as measured by densitometry using Image studio software ($n = 3$ independent experiments). **d** p62$^{fl/fl}$ MEFs cells infected with GFP or CRE adenoviruses and stably expressing Flag-p62 WT or D129K, were deprived of amino acids for 4 h and then stimulated with arginine for the indicated durations. Cell lysates were analyzed for the levels of the specified proteins. Graphs represent p-S6K/actin ratio as measured by densitometry using Image studio software ($n = 3$ independent experiments). Two-way ANOVA test **$p < 0.01$, ***$p < 0.001$. Error bar in **c** and **d** represents s.e.m. with N of 3

unable to bind arginine (D129K, D147K, or D149K) and analyzed the arginine-mediated mTORC1 activation under arginine (Fig. 5c) or amino acids (Fig. 5d) starvation. We found that overexpression of the p62 mutants, impaired in arginine binding, rescued the mTORC1 activation almost at the same level as WT p62. These data demonstrate that activation of the mTORC1 pathway by arginine is largely independent of the arginine-binding capacity of p62. Interestingly, the autophagy pathway was activated after starvation as indicated by notable lipidation of LC3 (compare LC3-II bands in Figs. 5c, d). Re-addition of arginine or amino acids had little effect (if not inhibiting) on autophagy under tested conditions. Taken together, our findings suggest that

unlike arginylated substrates, a free arginine is incapable of inducing autophagy.

**p62$_{ZZ}$ interacts with RL**. Modeling peptide mimetics in the RE-binding site of p62$_{ZZ}$ suggested that the sequence containing a tandem of positively charged lysine or arginine residues followed by a negatively charged glutamate or aspartate can mimic the Nt-RE sequence (Fig. 6). Intriguingly, the 20-residue linker connecting the PB1 and ZZ domains in p62 contains two such motifs, i.e., KKE (aa 102–104 of p62) and RRD (aa 106–108 of p62) (Fig. 6b). To determine whether p62$_{ZZ}$ can bind to the internal

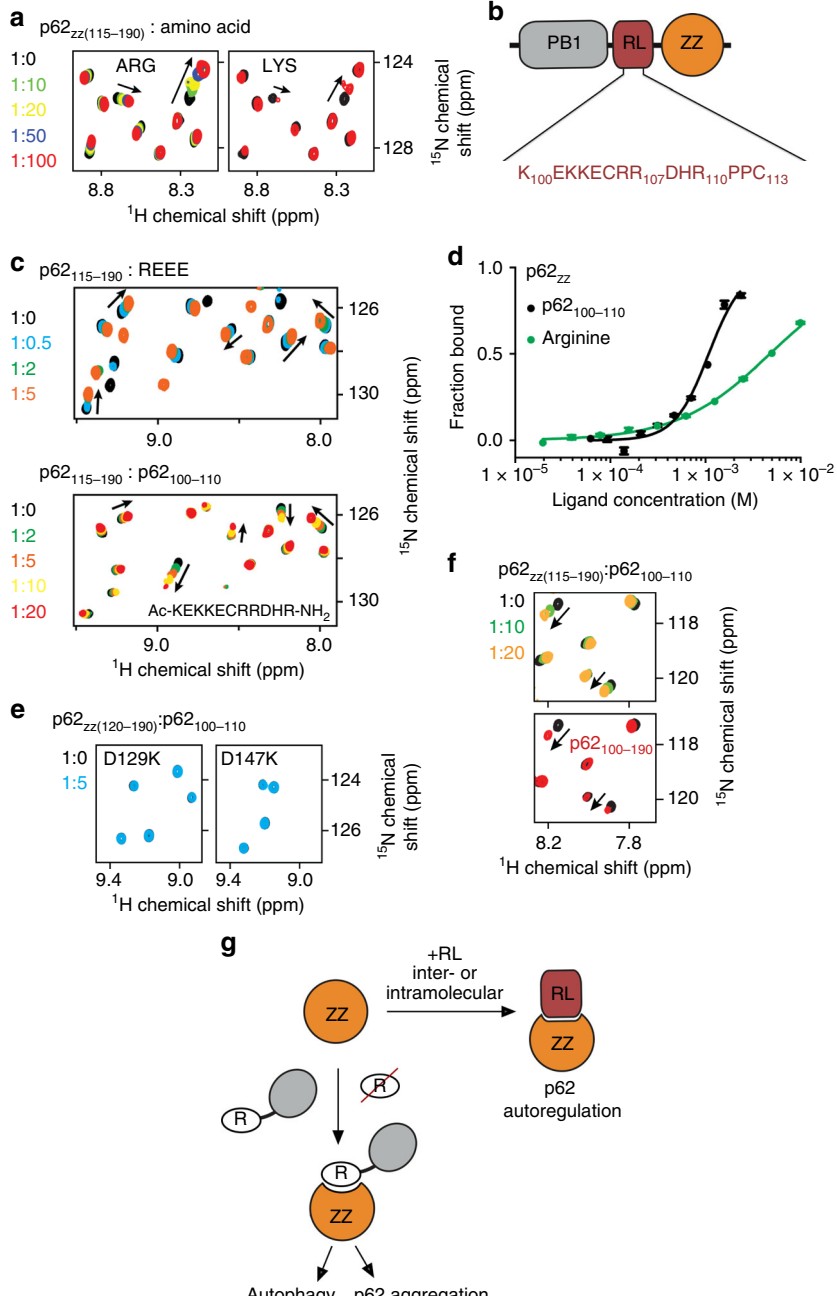

**Fig. 6** p62zz interacts with RL. **a** Superimposed $^1H,^{15}N$ HSQC spectra of p62$_{ZZ\ (115-190)}$ collected upon titration with the indicated amino acids showing that p62$_{ZZ}$ has a weak binding activity for lysine. Spectra are color coded according to the protein:ligand molar ratio. **b** Schematic representation of the p62 N-terminal region containing PB1 and ZZ domains. **c** Superimposed $^1H,^{15}N$ HSQC spectra of p62$_{ZZ\ (115-190)}$ collected upon titration with the indicated peptides. Spectra are color coded according to the protein:peptide molar ratio. **d** Binding curves used to determine binding affinities of p62$_{ZZ}$ by MST. Error bar represents s.d. from triplicate measurements. **e** Superimposed $^1H,^{15}N$ HSQC spectra of p62$_{ZZ\ (120-190)}$ collected upon titration with the RL peptide. Spectra are color coded according to the protein:peptide molar ratio. **f** Superimposed $^1H,^{15}N$ HSQC spectra of p62$_{ZZ\ (115-190)}$ collected upon titration with the p62$_{ZZ\ (100-110)}$ peptide (top) and superimposed $^1H,^{15}N$ HSQC spectra of p62$_{ZZ\ (115-190)}$ and p62$_{RL-ZZ\ (100-190)}$. **g** A model of p62 function and autoregulation. The ZZ domain interacts with the RL region either intra- or inter- molecularly. Binding of the ZZ domain to Nt-R degron but not arginine is necessary for p62 aggregation and autophagy

sequence, we probed association of p62$_{ZZ}$ with the K$_{100}$EKKECRRDHR$_{110}$ (p62$_{100-110}$) peptide by NMR and MST. Large CSPs in p62$_{ZZ}$ upon gradual addition of the p62$_{100-110}$ peptide indicated direct interaction (Fig. 6c). Although patterns of CSPs induced by REEE and p62$_{100-110}$ peptides were not identical, a similar set of amides was perturbed in either titration, implying that the same binding site of p62$_{ZZ}$ accommodates both peptides. In contrast, the D129K and D147K mutants of p62$_{ZZ}$,

which are defective in binding to Nt-R, did not interact with RL (Fig. 6e), corroborating the notion that the RL-binding site of p62$_{ZZ}$ at least partially overlaps with the Nt-R-binding site of this domain. MST measurements yielded a $K_d$ of 320 μM for the interaction between p62$_{ZZ}$ and the p62$_{100-110}$ peptide, however, we note that this interaction must be tighter in the physiologically relevant condition, because RL and ZZ are physically linked in p62 (Fig. 6d). Indeed, NMR amide resonances of the construct

containing RL natively linked to p62$_{ZZ}$ suggested formation of the stable RL-p62$_{ZZ}$ complex (Fig. 6f and Supplementary Figure 5).

In conclusion, in this study we have elucidated the molecular mechanism for selective recognition of the arginylated substrates by p62$_{ZZ}$. We show that binding of p62$_{ZZ}$ to Nt-R degron induces autophagy and that this interaction is required for p62 puncta formation in cells. We further demonstrate that p62 is necessary for arginine-mediated mTORC1 activation but p62 does not serve as a direct arginine sensor. We identified a regulatory linker (RL) between the PB1 domain and the ZZ domain that is recognized by p62$_{ZZ}$ in vitro. Although further studies are necessary to fully understand the interplay between the multiple binding partners of p62$_{ZZ}$, the data presented here suggest that recognition of the internal p62 sequence by p62$_{ZZ}$ could modulate p62 activities (Fig. 6g). Our results show that p62$_{ZZ}$ binds tighter to RL than to free arginine amino acid ($K_d = 1.5$ mM, Fig. 6d), however, weaker to Nt-R degron. This difference in binding affinities can be critical in directing p62 toward the distinct autophagic and mTORC1 signaling pathways. The autoregulation involving RL could in turn be modulated by p62 phosphorylation, particularly of the p62$_{ZZ}$ residues Tyr148 and Ser170. Future studies will be required to test this idea and to establish whether the p62$_{ZZ}$-RL interaction occurs in an intermolecular or intramolecular manner. Furthermore, it will be important to investigate the role of the disease-relevant mutations in p62$_{ZZ}$ and RL, for example, R107W and D129N, identified in patients with amyotrophic lateral sclerosis[28].

## Methods

**Protein expression and purification**. The human p62 ZZ domain (aa 100–190, aa 115–190, and aa 120–190) was cloned into a pGEX 6p-1 vector and expressed in BL21 (DE3) RIL cells. The p62 ZZ domain (aa 120–171) and RE-linked ZZ domain (aa 120–171 of p62, following the sequence RELGS) were cloned into a pCIOX vector with an N-terminal His-tag and Ulp1 cleavage site. Primers are listed in Supplementary Figure 6. Protein production was induced with 0.2 mM isopropyl β-D-1-thiogalactopyranoside (IPTG) overnight at 16 °C in Luria broth (LB) or minimal media (M9) supplemented with $^{15}$NH$_4$Cl and 0.05 mM ZnCl$_2$. The glutathione S-transferase (GST)-tagged ZZ proteins were purified on glutathione Sepharose 4B beads (GE Healthcare) in 20 mM Tris-HCl (pH 7.0) buffer, supplemented with 100 mM NaCl and 5 mM DTT. The GST tag was cleaved with PreScission protease overnight at 4 °C. The His-tagged proteins were purified on Ni-NTA beads (Qiagen) in 20 mM Tris-HCl (pH 7.5) buffer, supplemented with 300 mM NaCl and 10 mM β-mercaptoethanol. The His tag was cleaved overnight at 4 °C with PreScission or Ulp1 protease. Unlabeled and $^{15}$N-labeled proteins were further purified by size exclusion chromatography and concentrated in Millipore concentrators. All mutants were generated by site-directed mutagenesis using the Stratagene QuikChange mutagenesis protocol, grown, and purified as WT proteins.

**NMR experiments**. NMR experiments were carried out at 298 K on Varian INOVA 600 and 900 MHz spectrometers. NMR samples contained 0.1 mM uniformly $^{15}$N-labeled WT or mutated p62$_{ZZ}$ in 20 mM Tris-HCl (pH 7.0) buffer supplemented with 100 mM NaCl, 5 mM DTT, and 8% D$_2$O. Binding was characterized by monitoring CSPs in the proteins induced by peptides (synthesized by SynPeptide or KE BioChem) and amino acids. $K_d$ was determined by applying principal component analysis to the $^1$H, $^{15}$N HSQC titration spectra in TREND[29,30]. Each binding isotherm was fitted using the following equation:

$$p_{bound} = \|PC1\| = \frac{\left( ([L]+[P]+K_d) - \sqrt{([L]+[P]+K_d)^2 - 4[P][L]} \right)}{2[P]} \quad (1)$$

where $[L]$ is concentration of the ligand, $[P]$ is concentration of ZZ, $p_{bound}$ is the fraction of protein bound to ligand, and $\|PC1\|$ is the normalized principal component, obtained by TREND, which indicates the change in the population of the bound state. The errors of the $K_d$ value are the fitting uncertainties from nonlinear least-squares fits using Kaleidagraph.

**X-ray crystallography**. The RE-linked p62$_{ZZ}$ (aa 120–171 of p62, following the sequence RELGS) and p62$_{ZZ}$ (aa 120–171) in complex with arginine were crystalized. RE-p62$_{ZZ}$ was purified by size exclusion chromatography using a S100 (GE Healthcare) column equilibrated in buffer containing 20 mM Tris-HCl (pH 7.5), 100 mM NaCl and 1 mM TCEP and concentrated to 1.5–2.5 mg/mL. Crystals were obtained using hanging drop vapor diffusion against an equal volume of the well

solution containing 0.1 M BICINE, pH 8.0, 20% PEG 6000 at 18 °C. Crystals were cryoprotected with the addition of 30% glycerol to the well solution and the X-ray diffraction data were collected on the UC Denver X-ray crystallography core facility Rigaku Micromax 007 high-frequency microfocus X-ray generator equipped with a Pilatus 200 K 2D area detector. For crystallization of p62$_{ZZ}$ in complex with arginine, the eluted protein was precipitated by ultracentrifugation and then resolved in buffer containing 0.1 M arginine (pH 7.5) to a final concentration of 20 mg/mL. Crystals were obtained by seeding at 18 °C in condition containing 0.1 M Bicine (pH 9.0), 14% PEG 20 K, 4% 1,4-Dioxane. Data collection was performed at the ALS 4.2.2 beamline, Berkeley. Indexing, integrating, and scaling were processed by HKL3000[31]. The phase solution was found by single-wavelength Anomalous Dispersion method with Zn anomalous signal or using molecular replacement. Model building and refinement were carried out with Coot[32] and Phenix.refine[33]. The final structure was validated with MolProbity. The statistics of the data collection and refinement are summarized in Supplementary Table 1.

**Fluorescent MST-binding assay**. The MST experiments were performed using a Monolith NT.115 instrument (Nanotemper). All experiments were performed with the purified ZZ domain (aa 115–190) in a buffer containing 10 mM HEPES (pH 7.4), 150 mM NaCl and 1 mM TCEP. The final concentration of the fluorescein (FAM)-labeled peptide (KE BIOCHEM) was kept at 80 nM. Dissociation constants for the interaction between ZZ with unlabeled peptides (REEE, RAEE, and p62$_{100–110}$) and free arginine were measured using a displacement assay in which increasing amount of unlabeled peptides were added into a preformed ZZ:FAM-peptide complex prepared by supplementing 40 μM ZZ into each sample. The measurements were performed at 50% LED and 40% MST power with 3-s laser-on time and 25-s off time. For all measurements, samples were loaded into premium capillaries and 1300–1700 counts were obtained for the fluorescence intensity. The $K_d$ and IC$_{50}$ values were determined with the MO. Affinity Analysis software (NanoTemper Technologies GmbH). The $K_i$ values for unlabeled peptides with ZZ were determined from the IC$_{50}$ values observed in the displacement assay and converted by the following equation:

$$K_i = [I]_{50} / \left( \frac{[L]_{50}}{K_d} + \frac{[P]_0}{K_d} + 1 \right) \quad (2)$$

where $[I]_{50}$ is the concentration free unlabeled ligand at 50% binding, $[L]_{50}$ is the concentration of free labeled H3 peptide at 50% binding. The $K_d$ value is the dissociation constant of FAM-labeled peptide determined in the direct binding experiment described above. All measurements were done in triplicates.

**Fluorescence spectroscopy**. Spectra were recorded at 25 °C on a Fluoromax-3 spectrofluorometer (HORIBA). The samples containing 1.0 μM p62 ZZ domain (aa 120–171) and progressively increasing concentrations of the peptide were excited at 274 nm. Experiments were performed in buffer containing 10 mM HEPES (pH 7.4), 150 mM NaCl and 1 mM TCEP. Emission spectra were recorded over a range of wavelengths between 285 and 315 nm with a 0.5 nm step size and a 1-s integration time and averaged over three scans. The $K_D$ values were determined using a nonlinear least-squares analysis and the equation:

$$\Delta I = \Delta I_{max} \frac{\left( ([L]+[P]+K_d) - \sqrt{([L]+[P]+K_d)^2 - 4[P][L]} \right)}{2[P]} \quad (3)$$

where $[L]$ is the concentration of the peptide, $[P]$ is the concentration of ZZ domain, $\Delta I$ is the observed change of signal intensity, and $\Delta I_{max}$ is the difference in signal intensity of the free and bound states of the ZZ domain. The $K_D$ value was averaged over three separate experiments, with error calculated as the standard deviation between the runs.

**Amino acids and arginine starvation and arginine stimulation**. For amino acids starvation, MEFs[18] were rinsed with phosphate-buffered saline (PBS) and incubated in amino acid-free RPMI (USBiological), supplemented with 10% dialyzed serum, for 4 h. For arginine starvation, MEFs were rinsed with PBS and incubated with arginine, leucine and lysine-free RPMI (USBiological), supplemented with 0.38 mM of leucine and 0.21 mM of lysine, and 10% dialyzed serum, for 4 h . Cells were stimulated with 1 mM of arginine for different durations. Cells were rinsed with PBS and immediately lysed with Ripa lysis buffer (1% Triton, 0.5% SDS, 10 mM β-glycerol phosphate, 10 mM pyrophosphate, 40 mM Hepes pH 7.4, 2.5 mM MgCl$_2$ and 1 tablet of EDTA-free protease inhibitor (Roche)). The cell lysate was cleared by centrifugation at 13,000 rpm at 4 °C in a microcentrifuge for 10 minutes. Cell extracts were denatured by addition of 1× sample buffer followed by boiling for 5 min, resolved by 8–14% SDS-PAGE, and then transferred to nitrocellulose-ECL membranes (GE Healthcare).

**Measurement of autophagic flux by ratiometric flow cytometry**. Cells stably expressing mCherry-GFP-LC3 or mCherry-GFP-Fis1[34] were used for flow cytometric analysis. Briefly, a Gallios 561 (Beckman Coulter) using 488 and 561 nM lasers for green and red fluorophore excitation, respectively, was used to perform

flow cytometry. The appropriate forward/side scatter profile was used to exclude non-viable cells. Cells undergoing autophagy were defined as those expressing a high mCherry/GFP fluorescence ratio, as delivery by autophagy to the lysosome quenches the GFP signal, but not the mCherry signal. The gates to define what constituted an increased mCherry/GFP fluorescence ratio were set based on cells treated overnight with bafilomycin A1 (10 nM, Sigma-Aldrich B1792; CAS RN: 88899-55-2), a condition that represents cells with little or no autophagic flux. The bottom of the gate for each set of flow cytometry experiments was therefore set at the rightward base of the bafilomycin A1-treated curve with the gate such that no >5% of bafilomycin A1-treated cells were included in the gate.

**X-peptide pulldown assay**. A set of synthetic 12-mer peptides were C-terminally biotin conjugated. The X-nsP4 peptide (X-IFSTIEGRTYK-biotin) has Arg (arginylated) or Gly (stabilized) at the N-terminus.

A set of 10-mer BiP-derived peptides (X-EEDKKEDVG-biotin) and 10-mer CRT peptides (X-EPAVYFKEQ-biotin) bearing N-terminal Arg-Glu (permanently arginylated), Glu (native), or Val (glutamic acid-to-valine mutant) were used. For cross-liking with resin, C-terminally biotin-conjugated peptides were mixed with high capacity streptavidin agarose resin (Thermo, #20361) with a ratio of 0.5 mg peptide per 1 ml settled resin and incubated on rotator at 4 °C for overnight. After washing five times with PBS, the peptide–beads conjugates were diluted with PBS at 1:1 ratio. To prepare protein extracts, cells were collected by centrifugation and lysed by freezing and thawing at least ten times in Hypotonic Buffer (10 mM KCl, 1.5 mM MgCl$_2$, and 10 mM HEPES, pH 7.9) with a protease inhibitor mix (Sigma, P8340). After spinning down with centrifugation in 12,000 rpm at 4 °C for 20 min, proteins were quantified using BCA protein assay kit (Thermo Scientific, #23227). Total proteins (70 µg) diluted in 250 µl Binding Buffer (0.05% Tween-20, 10% glycerol, 0.2 M KCl, and 20 mM HEPES-pH 7.9) were mixed with 50 µl peptide-bead resin and incubated at 4 °C for 2 h on a rotator. The protein-bound beads were collected by centrifugation at 5000 rpm for 2 min and washed five times with binding buffer. The beads were resuspended in 25 µl SDS sample buffer, heated at 95 °C for 5 min, and subjected to SDS-PAGE and immunoblotting.

**In vitro p62 aggregation assays**. Cells were lysed by 10 cycles of freezing and thawing in lysis buffer (50 mM HEPES, pH 7.4, 0.15 M KCl, 0.1% Nonidet P-40, 10% glycerol, and a mixture of protease inhibitors and phosphatase inhibitor), followed by centrifugation at 13,000 × g for 20 min at 4 °C. The protein concentration in supernatant was determined using BCA assay. Total 5 µg protein was mixed with 50 mM of Arg-Ala or Ala-Arg dipeptide in the presence of 100 µM bestatin at room temperature for 2 h. Non-reducing sampling buffer containing 4% lithium dodecyl sulfate was added to each sample, heated at 95 °C for 2 min, and subjected to SDS-PAGE and immunoblotting.

**Cell culture and immunoblotting**. HEK293 and MEF cell lines were cultured in Dulbecco's modified Eagle's medium (GIBCO) supplemented with 10% fetal bovine serum (GIBCO) in a 5% CO$_2$ incubator. For immunoblotting, cells were lysed in SDS-based sample buffer (277.8 mM Tris-HCl, pH 6.8, 4.4% LDS, 44.4% (v/v) glycerol) with beta-mercaptoethanol. Using SDS-PAGE, whole-cell lysates were separated and transferred onto polyvinylidene difluoride membranes at 100 V for 2 h at 4 °C. Subsequently, the membrane was blocked with 4% skim milk in PBS solution for 30 min at room temperature and incubated overnight with primary antibodies, followed by incubation with host-specific horseradish perox-idase (HRP)-conjugated secondary antibodies. The primary antibodies used were: anti-phospho-S6K (Cell Signaling, #9205), anti-phospho-4EBP1 (Cell Signaling, #2855), anti-phospho-S6 (Cell Signaling, #5364), anti-LC3 (Cell Signaling #4108, Sigma #L7543, and Novus #NB100-2220), anti-β-actin (Sigma, #A1978 and #A5441), anti-Flag (Sigma #F3165 and Sigma #F1804), anti-p62 (Novus #H00008878 and Abcam #56416), and anti-GAPDH (BioWorld, #AP0063). The secondary antibodies used were: anti-IgG1 (BD Biosciences #550331, Cell Signaling #7974 and #7076), anti-rabbit (Dako, #E0432), and Alexa fluor488 anti-IgG (Invitrogen #A11034 and #11029). Details of antibodies and dilutions are listed in Supplementary Data 1. For signal detection, a mixture of ECL solution (Thermo Fisher Scientific) was applied onto the membrane and captured using X-ray films. Densitometry of band intensity was performed using ImageJ (NIH, Bethesda).

**Immunocytochemistry**. To observe cellular localization of proteins, cells were cultured on cover slips coated with poly-L-lysine. The cells were fixed with 4% paraformaldehyde in PBS (pH 7.4) for 15 min at room temperature and washed three times with PBS for 5 min. The cells were permeabilized with 0.5% Triton X-100/PBS solution for 15 min and washed three times with PBS for 5 min. The cells were blocked with 2% bovine serum albumin (BSA)/PBS solution for 1 h at room temperature. After blocking, the cells were incubated overnight at 4 °C with primary antibody diluted in 2% BSA/PBS solution. Following incubation, the cells were washed three times for 10 min with PBS and incubated with secondary antibody diluted in 2% BSA/PBS for 30 min at room temperature. Subsequently, the cover slips were mounted on glass slides using a DAPI-containing mounting medium. Confocal images were taken by laser scanning confocal microscope 510 Meta (Zeiss) and analyzed by Zeiss LSM Image Browser (ver. 4.2.0.121). Subsequently, cells were deemed to exhibit significant colocalization if >10 puncta

structures of the respective proteins showed association or full colocalization. Quantification results are shown as mean ± S.D. of three independent experiments.

## Data availability
The atomic coordinates and structure factors of the p62$_{ZZ}$ complexes have been deposited in the Protein Data Bank under the accession codes 6MIU and 6MJ7. Other data are available from the corresponding authors upon reasonable request.

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

## Acknowledgements

We thank Jay Nix at Beamline 4.2.2 of the ALS in Berkeley for help with X-ray crystallographic data collection and Ian Ganley for sharing the mCherry-GFP-Fis1 tandem construct. This work was supported by grants from NIH GM106416, GM101664 and GM100907 to T.G.K., CA192642 and CA218254 to M.T.D-M., DK108743 and CA211794 to J.M., CA150925 and CA190170 to A.T., and CA204020 to X.S., and by the Basic Science Research Programs of the National Research Foundation funded by the MSIP of Korea (NRF-2016R1A2B3011389 to Y.T.K.), Protech Inc. internal fund to Y.T.K., and the Brain Korea 21 PLUS Program to Y.T.K.

## Author contributions

Y.Z., S.R.M., J.F.L., J.W.A., C.G.T., C.H.J., B.E.F., M.R.H. and W. M. performed experiments and together with X.S., J.M., A.T., M.T.D-M., Y.T.K. and T.G.K. analyzed the data. Y.Z., M.T.D-M., Y.T.K. and T.G.K. wrote the manuscript with input from all authors.

## Additional information

**Competing interests:** The authors declare no competing interests.

