## [Peer Review File · Nature Communications]

Reviewers' comments:

Reviewer #1 (Remarks to the Author):

The manuscript by Zhang et al. determined the crystal structure of p62zz fused with an Nt-R substrate and revealed the recognition mode of Nt-R by p62zz at atomic resolution. Using NMR spectroscopy and biophysical assays, the authors confirmed that the interactions observed in the crystal are actually important for those in solution. Moreover, *in vivo* analyses revealed that the recognition of Nt-R substrates by p62zz is important for p62 body formation and autophagy that are induced by XIE62-1004 or MG132. Finally, the authors found that the RL region between PB1 and zz of p62 has a sequence similar to Nt-R substrates and binds to p62zz, which inhibits oligomerization of p62 and p62-induced autophagy. Based on these observations, the authors proposed a model that binding of Nt-R substrates to p62zz releases the auto-inhibited conformation of p62 and induces p62 oligomerization, which triggers p62-mediated autophagy. Throughout the manuscript, structural studies are fine and clearly explain how p62 ZZ recognizes the Nt-R substrates. The model of p62 autoinhibition and its release by Nt-R substrates is attractive and reasonable. These new knowledges will be beneficial for the field. However, there are some critical concerns in some experiments described below, which must be resolved prior to be published in Nature Communications.

Major points

- 1) The experiments of p62 puncta formation in Figure 3a,b are confusing. In the control image, the number of p62-D129Amyc puncta seems to be much larger than that of p62-FLmyc, why? Does the quantified graph in Figure 3b represent the ratio of cells with/without p62 puncta? If so, the ratio seems to represent the transfection efficiency of p62. All the cells expressing sufficient p62 mutant seem to have many puncta in both control and XIE62-1004 in Figure 3a. Perform the same experiments using stably expressing cells.
- 2) Measurement of autophagy activity in Figure 3 has not been properly performed. The ratio of LC3-II/LC3-I is NOT a gold standard for autophagosome formation. It is not clear whether the high LC3-II/LC3-I ratio represents increased formation of autophagosomes or inhibition of later steps of autophagy (for example, dysfunction of lysosomes). The turnover of LC3-II should be monitored in order to measure autophagy activity.
- 3) In Figure 5f, p62 mutants were also treated with Arg-Ala. However, if the authors want to propose the model in Figure 5g, they should show that these mutants of p62 could form oligomers irrespective of Arg-Ala. Perform the experiment with and without Arg-Ala.
- 4) In page 5, bottom, "binding of the RAEE peptide was reduced ~3-fold ... binding affinities were corroborated by intrinsic tyrosine fluorescence measurements (Fig. 2d,e)"
The ratio of 14uM/5.6uM is 2.5, and intrinsic tyrosine fluorescence measurements give the ratio of 3.2uM/1.9uM=1.7. From these two data, the authors cannot say that RAEE peptide has ~3-fold reduced affinity than REEE. In order to say that "the presence of a glutamic acid next to the arginine further enhances this interaction" in page 6, line124, more reliable data are required.
- 5) In Figure 5f and 5g, what is the role of Cys113 and disulfide bond? Is Cys113 necessary for p62 oligomer formation? Perform the experiments with and without reducing agents.

Minor points

- 1) In Figure 2e, describe the meaning of superscripts a,b and c in Figure legend.
- 2) In Figure 2f and 2g, why did the band of pulled-down p62WT show a little slow migration compared with input bands?

Reviewer #2 (Remarks to the Author):

1. This is a well written paper providing some new structural insights in to the role of the ZZ-

domain of p62 in regulating autophagy. They make an ArgGlu – flexible linker – ZZ construct and use this to generate a crystal structure, this seemingly works nicely. Did the authors attempt to crystallise with peptide fragments rather than conjugate without success?

2. The authors characterise ZZ/ArgEEE interaction by MST – an affinity of 5.6 μM ; was this corroborated with other data, e.g., ITC or from the NMR titration experiments? They come to the conclusion that the p62 ZZ requires an acidic residue in position 2 and that this contributes to N-end selectivity as UBR-boxes prefer a hydrophobic, although the 3-fold decrease in affinity is a modest effect.

3. In isolation p62 has very weak affinity for Arg (1.5 mM), and so probably isn't a sensor for amino acid starvation. Did they characterise the affinity with the C-terminal amide analogue of free Arg – presumably the C-terminal carboxylate has a key role in modulating the (low) affinity of free Arg?

4. The conclusion that a sequence of basic/basic/acidic residues would be recognised by ZZ-domain, and that the occurrence of two of these motifs in the PB1 linker is interesting. So they generate a peptide for the linker and test the interaction by MST. They demonstrate that actually p62 has a weak affinity for this peptide ($\sim 320 \mu\text{M}$). They propose the model that perhaps the ZZ and bridge region interact under 'basal' conditions, and that high affinity substrates compete for binding. Although a plausible model, it seems to contradict previously published work whereby the oligomerisation drives activation. Perhaps the authors would like to comment.

5. All in all the structural work is interesting, but they repeat a number of assays (e.g. X1E62-1004 treatment and Arg-Ala mediated oligomerisation) previously reported in three papers from the Korean group (one in Nature Comms), making this study slightly incremental.

6. In Fig 1d,e and in Fig 5a,c it would be helpful if the residues that are perturbed in the NMR titrations were specifically labelled.

Reviewer #3 (Remarks to the Author):

In their manuscript Zhang et al describe a new study that builds on the previous ones describing the multifunctional adaptor protein p62/SQSTM1 as an N-recognin for the N-end rule pathway and in particular for proteins having an Arg residue at their N-terminus. The new data presented herein concern the structural features of the ZZ domain of p62, revealing the mode of binding between ZZ and Arg-Glu and critical residues involved (such as D129, D147, and D149 of ZZ) in the interaction. The authors determine selectivity features of the ZZ:Arg-Glu binding and characterise affinities using relevant biophysical methods, such as NMR. To demonstrate the relevance of the discovered structural ZZ features for the p62 biology, Zhang et al utilise D129K, D147K, and D149K mutants of p62-ZZ, which fail to bind Arg in the Arg-Glu dipeptide in biophysical assays, and show that the p62 mutants cannot form aggregates in response to X1E62-1004 and MG132 (treatments that the authors showed previously to potentially induce p62 aggregation and autophagy in cells; Cha-Molstad et al. 2017 Nat Comm). Of interest is the finding that free Arg has very low affinity to the ZZ domain of p62 and does not modulate p62 aggregation and/or p62-mediated mTOR activation. Of potentially high significance is the hypothesis that the authors put forth about the intramolecular interaction of the ZZ domain with a short linker (aa 102-108 in human p62, KKECRRDH) that contains stretches of both Arg and Lys. The authors use NMR to demonstrate binding between the ZZ and an isolated peptide matching the 100-110 aa sequence of p62 and then go on to show that D129K, D147K, and D149K p62 mutants (which would fail to bind this stretch) are oligomeric in contrast to the wild-type (WT) p62. The authors suggest that the closed conformation of p62 is maintained by ZZ bound weakly to the EKKECRRDHR linker (Kd of ca. 300 μM). This closed conformation can be released upon higher affinity interaction with arginylated

proteins (mimicked by Arg-Ala in their assays, Kd of ca. 14 μ M).

I find this study of high relevance for several fields of molecular biology (protein degradation, autophagy, protein domain functions). It is accurately executed, especially in the areas of structural biology and biophysics. The biological part is a bit weak, and some of the data around the induction of the p62 aggregates by XIE62-1004 and MG-132 have already been published by the same group. So, the manuscript would require some additional mechanistic and/or functional data to claim a major advancement over the already published material. I have a few suggestions which might help improve this otherwise very interesting work.

Specific points:

1. The model of autoinhibitory engagement of ZZ with the EKKECRRDHR linker

I find the authors' hypothesis on the intramolecular autoinhibition of p62 interesting but not entirely convincing. Is there a way to demonstrate that the molecule indeed opens using a FRET assay? Further, is that so clear that the ZZ:EKKECRRDHR interaction is occurring in cis rather than in trans with another p62 molecule? If it is an inter- rather than intramolecular interaction, the role of the PB1 domain in the whole model might be much more relevant. The authors published previously that PB1 was absolutely required for p62 aggregation and yet they do not address its role in this study. Can the PB1 D69A construct still aggregate upon treatment with compounds that outcompete ZZ:EKKECRRDHR interaction due to their higher binding affinity (i.e. Arg-Ala or XIE62-1004). A minor point: the authors do not show that D129K, D147K, and D149K p62 mutants are oligomeric in the absence of the Arg-Ala reagent (Fig. 5f).

2. Role of NBR1, the binding partner of p62

NBR1 interacts with p62 via the PB1 domain, and it is therefore difficult to study p62 oligomerisation when NBR1 (which is also prone to self-oligomerisation via its coiled-coil domains and via p62 interaction) is present in the cells. What would be the result of some of the authors' present studies in NBR1 KO/KD cells? Further, NBR1 has a conserved ZZ domain, which lacks the p62-specific Asp residues. The authors should in the least comment on this interesting feature in the discussion part of their manuscript.

We thank the Editor and Reviewers for the insightful and very constructive comments, which were helpful in revising and strengthening this manuscript.

Reviewer 1, Comment 1: The experiments of p62 puncta formation in Figure 3a,b are confusing. In the control image, the number of p62-D129Amyc puncta seems to be much larger than that of p62-FLmyc, why? Does the quantified graph in Figure 3b represent the ratio of cells with/without p62 puncta? If so, the ratio seems to represent the transfection efficiency of p62. All the cells expressing sufficient p62 mutant seem to have many puncta in both control and XIE62-1004 in Figure 3a. Perform the same experiments using stably expressing cells.

Author's response: As suggested, we have performed these experiments in mouse embryonic fibroblasts (MEFs) stably expressing p62-WT-flag or p62-D129K-flag (new Fig. 3a, b). The new data confirmed that the D129K mutant is incapable of puncta formation.

Reviewer 1, Comment 2: Measurement of autophagy activity in Figure 3 has not been properly performed. The ratio of LC3-II/LC3-I is NOT a gold standard for autophagosome formation. It is not clear whether the high LC3-II/LC3-I ratio represents increased formation of autophagosomes or inhibition of later steps of autophagy (for example, dysfunction of lysosomes). The turnover of LC3-II should be monitored in order to measure autophagy activity.

Author's response: As suggested we have monitored LC3-II turnover in new Fig. 3c and Suppl. Fig. S3.

Reviewer 1, Comment 3: In Figure 5f, p62 mutants were also treated with Arg-Ala. However, if the authors want to propose the model in Figure 5g, they should show that these mutants of p62 could form oligomers irrespective of Arg-Ala. Perform the experiment with and without Arg-Ala.

Author's response: We have performed the polymerization assays with and without Arg-Ala peptide (Suppl. Fig. S4). The new text has been added to the section "p62_{ZZ} mediates p62 aggregation *in vitro*" on page 9.

We have also revised the model (Fig. 5g) and the text in the sections "The p62_{ZZ} interacts with RL" (pages 10-11) and "Concluding remarks" (page 11).

Reviewer 1, Comment 4: In page 5, bottom, "binding of the RAEE peptide was reduced ~3-fold ... binding affinities were corroborated by intrinsic tyrosine fluorescence measurements (Fig. 2d,e)" The ratio of 14uM/5.6uM is 2.5, and intrinsic tyrosine fluorescence measurements give the ratio of 3.2uM/1.9uM=1.7. From these two data, the authors cannot say that RAEE peptide has ~3-fold reduced affinity than REEE. In order to say that "the presence of a glutamic acid next to the arginine further enhances this interaction" in page 6, line124, more reliable data are required.

Author's response: We agree, the sentence on page 5 has been revised to: In agreement, dissociation constant (K_d) measured by microscale thermophoresis (MST) for the interaction of p62_{ZZ} with REEE was found to be 5.6 μ M, whereas binding of the RAEE peptide was reduced ($K_d=14 \mu$ M) (Fig. 2c-e).

Reviewer 1, Comment 5: In Figure 5f and 5g, what is the role of Cys113 and disulfide bond? Is Cys113 necessary for p62 oligomer formation? Perform the experiments with and without reducing agents.

Author's response: As suggested we have performed experiments with and without beta-ME (please see Suppl. Fig. S4a). These data suggest that the disulfide bond formation is necessary

for p62 polymerization, confirming our previous mass spectrometry analysis showing that Nt-R induces polymerization of p62 to a high degree through Cys113 disulfide bond formation (Chamolstad et al, 2017).

Reviewer 1, Comment 6: Minor points

In Figure 2e, describe the meaning of superscripts a,b and c in Figure legend. – done, thank you.

In Figure 2f and 2g, why did the band of pulled-down p62WT show a little slow migration compared with input bands? - Input is original cell lysate, whereas bead-bound p62WT is a pulled-down and washed protein.

Reviewer 2, Comment 1: Did the authors attempt to crystallise with peptide fragments rather than conjugate without success?

Author's response: Although we produced many constructs of p62-ZZ and purchased several REX peptides, we were unable to co-crystallize them. We took advantage of the elegant approach developed in several X-ray crystallography labs that aids in crystallization of the complexes between interacting proteins and peptides by generating a linked chimeric construct (Chen et al., 2015; Wang et al., 2010).

Reviewer 2, Comment 2: The authors characterise ZZ/ArgEEE interaction by MST – an affinity of 5.6 μ M; was this corroborated with other data, e.g., ITC or from the NMR titration experiments?

Author's response: Yes, the MST data (Fig. 2c, e) have been corroborated by tyrosine fluorescence data (Fig. 2c, e) and NMR HSQC titrations (Fig.1d, e).

Reviewer 2, Comment 3: In isolation p62 has very weak affinity for Arg (1.5 mM), and so probably isn't a sensor for amino acid starvation. Did they characterise the affinity with the C-terminal amide analogue of free Arg – presumably the C-terminal carboxylate has a key role in modulating the (low) affinity of free Arg?

Author's response: We agree, and our data demonstrate that p62 is not a sensor for free amino acid Arg, however it is a naturally occurring state of arginine in cell, it does not normally exist as an amide.

Reviewer 2, Comment 4: ...Although a plausible model, it seems to contradict previously published work whereby the oligomerisation drives activation. Perhaps the authors would like to comment.

Author's response: We have performed additional assays that confirmed that autophagic p62 aggregation depends on both the functional ZZ domain and stimulation with Arg-Ala (Suppl. Fig. S4). We have also added new text to the section "p62_{ZZ} mediates p62 aggregation *in vitro*" on page 9, and revised the model (Fig. 5g) and the text in the sections "The p62_{ZZ} interacts with RL" (pages 10-11) and "Concluding remarks" (page 11).

Reviewer 2, Comment 5: All in all the structural work is interesting, but they repeat a number of assays (e.g. X1E62-1004 treatment and Arg-Ala mediated oligomerisation) previously reported in three papers from the Korean group (one in Nature Comms), making this study slightly incremental.

Author's response: It was necessary to perform a set of assays that were developed by the YTK's group to substantiate our structural studies. In addition, to further characterize the role of

p62, new data (Figs. 3c, 3e-i, 5e, f, and Suppl. Figs. S3, S4, S5) have been obtained and discussed in the revised manuscript.

Reviewer 2, Comment 6: In Fig 1d,e and in Fig 5a,c it would be helpful if the residues that are perturbed in the NMR titrations were specifically labelled.

Author's response: Because sequence specific resonance assignment for the p62 ZZ domain is not available, unfortunately we could not label crosspeaks.

Reviewer 3, Comment 1: The model of autoinhibitory engagement of ZZ with the EKKECRRDHR linker I find the authors' hypothesis on the intramolecular autoinhibition of p62 interesting but not entirely convincing. Is there a way to demonstrate that the molecule indeed opens using a FRET assay? Further, is that so clear that the ZZ:EKKECRRDHR interaction is occurring in cis rather than in trans with another p62 molecule? If it is an inter- rather than intramolecular interaction, the role of the PB1 domain in the whole model might be much more relevant. The authors published previously that PB1 was absolutely required for p62 aggregation and yet they do not address its role in this study. Can the PB1 D69A construct still aggregate upon treatment with compounds that outcompete ZZ:EKKECRRDHR interaction due to their higher binding affinity (i.e. Arg-Ala or XIE62-1004).

Author's response: We very much appreciate these suggestions, and we agree with the reviewer- it is very challenging to delineate cis/trans and inter- vs intra-molecular interactions. We have generated a new construct of p62 containing both RL and ZZ (aa 100-190) and superimposed HSQC spectrum of this new construct with the HSQC spectra of individual ZZ (aa 115-190) in the free state and bound to RL (aa 100-110). Structural analysis (Fig. 5f and Suppl. Fig. S5) suggests that the ZZ domain binds to the RL sequence either directly linked to ZZ or added as a separate peptide.

Re the role of PB1 in p62 oligomerization. Indeed, previous findings by the YTK group showed that PB1 is required for the formation of ~250 kDa oligomers (please see Figure on the left, which is adapted from Chamolstad et al, 2017), and our new data (Suppl. Fig. S4b, c) confirmed that this oligomerization step is ZZ-independent. However, p62 aggregation requires the functional ZZ domain. The new text has been added to the section "p62_{ZZ} mediates p62 aggregation *in vitro*" on page 9.

Reviewer 3, Comment 2: A minor point: the authors do not show that D129K, D147K, and D149K p62 mutants are oligomeric in the absence of the Arg-Ala reagent (Fig. 5f).

Author's response: Please see our response to comment 3 of Reviewer 1.

Reviewer 3, Comment 3: Role of NBR1, the binding partner of p62
NBR1 interacts with p62 via the PB1 domain, and it is therefore difficult to study p62 oligomerisation when NBR1 (which is also prone to self-oligomerisation via its coiled-coil domains and via p62 interaction) is present in the cells. What would be the result of some of the authors' present studies in NBR1 KO/KD cells? Further, NBR1 has a conserved ZZ domain, which lacks the p62-specific Asp residues. The authors should in the least comment on this interesting feature in the discussion part of their manuscript

Author's response: We thank the reviewer for this suggestion- we were interested in NBR1, however, our *in vitro* pulldown assays show that NBR1 does not bind to the Nt-R (unpublished

data), suggesting that NBR1 is not involved in the modulation of p62 oligomerization by N-end rule ligands.

Reviewers' comments:

Reviewer #1 (Remarks to the Author):

The authors have addressed all my concerns.

Reviewer #3 (Remarks to the Author):

I would like to thank Zhang et al for considerably improving the manuscript. The clarification of the role of PB1 vs ZZ domains in p62 oligomerisation vs aggregation, respectively, is very important. The clarification with respect to cis vs trans interaction between the ZZ domain and the EKKECRRDHR linker (neither can be ruled out at present) is of value too. Many thanks for commenting on the role of ZZ domain in NBR1 (unpublished data). While the manuscript has now gained more clarity and validity, I have to raise one remaining point that, in my opinion, is key for the claim that the authors make with respect to the role of p62 aggregates in instigating the autophagosome formation, or more broadly autophagy. In the Figure 3, the authors show that the Nt-R mimetic XIE62-1004 induces LC3 lipidation and enhances autophagic flux in p62-dependent manner, suggesting that p62 aggregation is indeed important for autophagy induction. Apparently, use of p62 mutants (D147K and D129K), which fail to bind the Nt-R and the EKKECRRDHR linker, cannot rescue the XIE62-1004 effect in p62^{-/-} cells. If the role of p62 aggregation is so critical in autophagy, it can be predicted that the aggregation-deficient mutant Cys113 (Cha-Molstad et al., 2017) would also fail to rescue the XIE62-1004 effect in p62^{-/-} cells. I am requesting these additional data to conclude this manuscript.

We thank the Editor and Reviewer3 for the insightful and very constructive comments, which were helpful in revising and strengthening this manuscript.

Comment 1: In the Figure 3, the authors show that the Nt-R mimetic XIE62-1004 induces LC3 lipidation and enhances autophagic flux in p62-dependent manner, suggesting that p62 aggregation is indeed important for autophagy induction. Apparently, use of p62 mutants (D147K and D129K), which fail to bind the Nt-R and the EKKECRRDHR linker, cannot rescue the XIE62-1004 effect in p62^{-/-} cells. If the role of p62 aggregation is so critical in autophagy, it can be predicted that the aggregation-deficient mutant Cys113 (Cha-Molstad et al., 2017) would also fail to rescue the XIE62-1004 effect in p62^{-/-} cells. I am requesting these additional data to conclude this manuscript.

Author's response: As suggested, we have performed additional experiments to confirm that aggregation of p62 via Cys113 is also essential for autophagy induction (please see Figure on the left). Treatment of p62^{+/+} cells with XIE62-1004 induced LC3-II lipidation (lanes 1 vs. 2), whereas p62^{-/-} cells were non-responsive (lanes 3 vs. 4). As expected, transient expression of recombinant p62-myc in p62^{-/-} MEFs rescued their ability to induce LC3-II lipidation upon XIE62-1004 treatment (lanes 5 vs. 6). Importantly, consistent with our current model, C113A mutant failed to induce autophagy upon XIE62-1004 treatment (lanes 7 vs. 8). Intriguingly, the mere expression of C113A mutant was sufficient to increase the level of LC3-II (lanes 5 vs. 7). While

further investigations are needed to differentiate the role of p62 in induced and non-induced autophagy, these data highlight that both Nt-R interaction with p62 and C113-dependent aggregation of p62 are important in induced autophagy.

REVIEWERS' COMMENTS:

Reviewer #3 (Remarks to the Author):

I thank the authors for the additional experiment and state that the authors have now addressed all my concerns. It is recommended that the new data be enclosed with the final version of the manuscript.